# Familial and Parental Predictors of Physical Activity in Late Adolescence: Prospective Analysis over a Two-Year Period

**DOI:** 10.3390/healthcare9020132

**Published:** 2021-01-29

**Authors:** Damir Sekulic, Dora Maric, Sime Versic, Ante Zevrnja, Admir Terzic, Natasa Zenic

**Affiliations:** 1Faculty of Kinesiology, University of Split, 21000 Split, Croatia; dado@kifst.hr (D.S.); simeversic@gmail.com (S.V.); 2PhD in Health Promotion and Cognitive Sciences, Sport and Exercise Sciences Research Unit, Department of Psychology Educational Science and Human Movement, University of Palermo, 90100 Palermo, Italy; dora.maric@unipa.it; 3Faculty of Medicine, University of Mostar, 88000 Mostar, Bosnia and Herzegovina; antezevrnja17@gmail.com; 4Clinical Hospital Split, 21000 Split, Croatia; 5Faculty of Physical Education and Sports, University of Tuzla, 75000 Tuzla, Bosnia and Herzegovina; admir.terza@bih.net.ba

**Keywords:** puberty, physical activity, familial factors, predictors, risk factors

## Abstract

Children’s health behaviors are highly influenced by their parents and family. This study aimed to prospectively evaluate the parental/familial factors associated with physical activity levels (PALs) among older adolescents. The participants were 766 adolescents, who were prospectively observed at baseline (when they were 16 years of age), at first follow-up measurement (FU1; 17 years of age), and second follow-up measurement (FU2; 18 years of age). Sociodemographic factors (age, gender, socioeconomic status, and sport participation) and parental/familial variables were evaluated at baseline. PALs (evidenced by the Physical-Activity Questionnaire-for-Adolescents) were prospectively evidenced at baseline, FU1, and FU2. Factorial analysis of variance for repeated measurements showed a significant decrease in PALs during the study course (F = 83.05, *p* < 0.001). Sport participation and male gender were significant predictors of PALs at baseline, FU1, and FU2. Logistic regression, controlled for sport participation and male gender, evidenced paternal education as a significant predictor of baseline PALs. Parental conflict was a significant predictor of PALs in all three testing waves. The significant influence of paternal education on the children’s PALs existed from younger adolescence until the age of 17 years. The association between parental conflict and PALs developed in older adolescence. These results should be used in the development of specific and targeted interventions aimed at the improvement of PALs and a reduction of sedentarism in youth.

## 1. Introduction

Despite the globally known benefits of physical activity (PA) and the negative health outcomes of an inadequate (PALs) physical activity levels, physical inactivity indicates a trend of pandemic proportions, with almost no improvement since 2001 [1]. Guthold et al. reported a slight decrease in boys insufficient PA since 2001, while there were no changes over time in girls, and in the general adolescent population [2]. Trends of PA decline in adolescence are repeatedly highlighting in Southeastern Europe as well [3,4,5,6]. It is evident that on-going technological development and economic growth have a strong influence on the global population lifestyle [7]. More precisely, changes in transport patterns (relying more on motorized transportation) and the transition toward sedentary occupations and leisure, which implies a more frequent use of technology for recreation and work, leads to an increase in sedentary behaviors and decrease in PALs across the world [1]. Additionally, advances in technology are changing the way modern-day youths think, play, socialize, and entertain, which often leads to the replacement of time spent engaging in PA with screen time [8].

The widespread problem of physical inactivity threatens the future of the world as we know it, especially considering the fact that 81% of adolescents (11–17 years) fail to meet the WHO recommendations of at least 60 min of moderate-to-vigorous-intensity physical activity per day [1]. This is particularly concerning, since an inadequate PAL is known to have a negative impact on health systems, economic progress, the environment, community well-being, and quality of life. Therefore, the question arises as to who will carry the future of the world if, according to the data, the younger generations will most likely pose a burden on the health systems and further economic development. Given the circumstances, in order to prevent negative outcomes and to change the negative trends in PALs, it is important to identify the social and environmental factors associated with PA, especially during adolescence [9]. 

Adolescence presents a period of many changes and has often been presented as a time of hardship and stress. It is a crucial period for interventions aiming to promote the greater involvement in some form of PA, leading to improvement of regular PA, which is known to have a positive impact on cardio-metabolic health (glucose and insulin resistance, blood pressure, and dyslipidemia), physical fitness (muscular and cardiorespiratory fitness), adiposity reduction, bone health, mental health (reduced symptoms of depression), and cognitive (executive function, academic performance) outcomes [1]. In general, adolescence is an important time in a young person’s life for the adoption and retention of future habits and for forming attitudes about various life matters such as health and educational habits. Therefore, it is reasonable to assume that physical activity during adolescence defines activity levels later in life, and thus directly affects an individual’s overall health status in the future [10,11]. 

Consequently, the factors associated with PALs in adolescence have been frequently explored. Faigenbaum et al. stated that ”physical activity is a learned behavior and is influenced by a variety of socio-ecological factors including, facilitators (e.g., friends, teachers, coaches, fitness specialists, and health care providers), facilities (e.g., schools, playgrounds, and sport fields) and family (e.g., parents and siblings)” [8]. Sallis et al. put forward sex, age (inverse relationship), ethnicity, depression (inverse relationship), perceived activity competence, previous physical activity, sibling PA, opportunities to exercise, parental support, and direct help from parents as the most consistent correlates of PA in a review of the literature. Although most studies show consistency when it comes to the impact of the abovementioned variables, many authors have pointed out the existence of studies with an inverse relationship or no impact of the discussed variables [12]. Furthermore, according to the vast majority of studies, participation in sport activities, level of education, self-efficacy, parental activity, and parental support are often positively correlated with PALs [13,14]. 

It is well established that families have a strong influence on children’s and adolescent’s health behavior, including PALs [12,15,16]. More precisely, parents (caretakers in general) are one of the primary influences of youth behavioral patterns, including PA-related actions (e.g., outdoor play, sport, and exercise) [12,17,18,19]. This is understandable if we take into consideration that most young people spend approximately 18 years of their life in close proximity to their parents [17,20]. During this period, parents usually serve as a model for future behavior and habits, directly influencing, in this case, PA behavior, although not always with conscious/deliberate intention [17,21]. According to the research conducted thus far, parental influence on PA can take many forms and, generally, can influence children’s PA, both directly and indirectly [17,22]. However, despite the wide consensus that parental factors are important determinants of children’s PALs, the results of studies thus far inconsistent. 

Gustafson and Rhodes, in their review, presented some inconsistent results; more specifically, six studies found a moderate correlation between parents’ PA and children’s PA, while seven studies did not support these findings. However, parental support was almost consistently recognized as a positive impact on children’s PALs, while this effect was more significant in younger children [23]. Similar outcomes were noted by Trost et al., who reported no support for the link between parents’ and children’s PA, while indicating a consistent positive association for parental support and children’s PA behavior [16]. Another systematic review, by Jaeschke et al., which reported on the determinants of PA, pointed out that encouragement from significant others was associated with greater PA in children and adolescents, whereas parental marital status and parental modeling were not associated with PA in children [24]. Furthermore, research examining parental and peer influence on older adolescents (17–19 years) found no association between adolescents’ and parents’ PA; however, in accordance with the previously mentioned studies, parents’ encouragement was related to PALs [25]. Parental education is inconsistently associated with young people’s PALs according to Sallis et al., which has also been reported in other studies [26]. A recent prospective study examining younger adolescents found a positive influence of paternal education level on PALs, yet no significant influence of maternal education, parental care, parental/familial conflict, parental absence from home, and parental questioning were found [3].

The previously presented inconsistencies in results are not surprising if we take into account the fact that previous studies were conducted on different populations varying in terms of socioeconomic status, ethnicity, and environmental characteristics. Besides this, they often differed in their measures of parental PA influence and modes for estimating PA levels. Moreover, it is important to emphasize the fact that there is a lack of information on changes in perception of parental support and influence on PA over time, since it has rarely been investigated. To acquire a clearer picture of these changes, especially regarding adolescents, longitudinal and systematic analysis of diverse factors is required. A prospective study would provide a clearer perspective and a more complete interpretation of the cause–effect relationship of the observed variables [13,24]. 

In a previously presented study, the authors provided prospective information about the influence of parental/familial variables on PALs in adolescents from Bosnia and Herzegovina between the age of 14 and 16 years [3]. In brief, parental education was significantly associated with PALs, but no association was found for the variables explaining parental/familial support and conflict with PALs in this period of life. For a more profound understanding of the problem, further analyses are needed in the in the next period of life (e.g., in older age). Therefore, this study aimed to prospectively determine the parental and familial factors associated with PALs and changes in PALs between 16 and 18 years of age. We were particularly interested in eventual changes in the relationships between variables that may occur in the observed period of life. Additionally, we evidenced changes in PALs that occurred in the study period. 

## 2. Materials and Methods

### 2.1. Design and Participants

Adolescents from three counties in Bosnia and Herzegovina participated in this prospective study. The research was conducted across three testing waves, with baseline testing at the beginning (September) of the 2017/2018 school year, when participants were approximately 16 years old and were starting the third grade of high school. The first follow-up testing occurred at the end of the 2017/2018 school year (May/June 2018), while the second follow-up testing was conducted at the end of high-school education (May/June 2019), when the participants were approximately 18 years old. The sampling was done across several phases. In the first step, all high schools in selected counties were grouped into two clusters according to their size (number of students). After this, we randomly selected one-half of the schools from each cluster, and finally, one-half of the third grades from the chosen schools. Considering that some high school programs last three years, in our study, we included only classes participating in four-year high school programs. A total of 856 participants were tested at the study baseline, but the final analysis included 766 participants (347 females) who carried out the testing on all three occasions (the drop-out rate was 15%). The sample used in this study met the theoretical required sample for the observed population in studied counties (341 required participants for a theoretical sample of approximately 3000 adolescents aged 16 years, and a level of significance of *p* < 0.05) (Figure 1).

### 2.2. Instruments

In both testing waves, the participants were examined with questionnaires that included sociodemographic characteristics, sport factors, parental/familial factors, and PALs. The questionnaires used were previously validated on a similar sample of participants, and details on testing are presented elsewhere [3,27].

The sociodemographic variables were age (in years) and gender (male and female). The parental/familial variables included: (i) Socioeconomic status (self-reported as below average, average, and above average), (ii) parental education level (elementary school, high school, and college/university degree—reported separately for mothers and fathers), (iii) conflict with parents (almost never, rarely, periodically, and often), (iv) parental presence at home (always at home, rarely absent, occasionally absent, and often absent), (v) evaluation of parental care (parents do not care at all, do not care enough, provide good care, and care very much), and (vi) level of parental questioning about school, friends, and similar problems (mostly never, rarely, from time to time, and often). The sport participation at baseline was evidenced by one question with three possible answers (participating, participated but quit, never participated—later categorized as participation vs. non-participation). 

PALs were evidenced at baseline, at first follow-up measurement (FU1), and at second follow-up measurement (FU2) by the Physical Activity Questionnaire for Adolescents (PAQ-A), which was previously used and validated with similar samples of participants [4,28,29]. This tool consists of nine questions that examine participants’ PALs over the previous seven days. While the last question is of an open type and serves only to record possible injuries and illness, the first eight relate to various types of PA (e.g., free play, sport at school, and sport at sport clubs), and all together, they form the final grade on a scale from 0 (minimum) to 5 (maximum PAL). Apart from the raw scores, for the purpose of this study, the results were categorized as previously suggested [3]. In brief, participants were stratified into groups with sufficient (score of 2.71 and higher) and insufficient (scores < 2.71) PALs. 

### 2.3. Procedures

Before conducting the testing, the examiners visited the schools and informed the students about the aims, purposes, and procedures of the examination. The students were given consent forms for participation in the study, and only those who brought back forms signed by their parents or guardians were included. The baseline testing was held two weeks after the first meeting. The questionnaires were completely anonymous, but because of study design (testing at three time points and the need to compare the results), students were instructed to use their private password for baseline and follow-up testing. The testing itself was conducted using an online survey platform, and participants filled in the questionnaires for approximately 15 min during school hours using their mobile phones. The study was approved by the Ethics Committee of the Faculty of Kinesiology, University of Split, and was completed in line with the proposed ethical guidelines.

The intra-cluster coefficient of 0.05 for the PALs at the baseline (with schools as clusters) indicated proper within-school variance. With regard to the drop-out rate, we found no significant differences in the baseline PALs between those adolescents who dropped out from the study and those who were tested at all three testing waves (*t*-test = 0.87, *p* > 0.05), but significantly more females were retained in the study (χ^2^ = 1.1, *p* < 0.05), which was attributed to the regularly reported more frequent absence from school among boys [30]. 

### 2.4. Data Analysis

The Kolmogorov–Smirnov test was used to test the normality of the distribution. Accordingly, the means and 95% confidence intervals (CI) were calculated for the PAQ-A, while for the other variables, the descriptive statistics included frequencies and percentages.

To estimate the differences in PALs among the testing waves, factorial analysis of the variance was used for the repeated measures (ANOVA), with “gender” and “measurement” as the main effects and interaction (gender × measurement). Additionally, consecutive Scheffe post-hoc tests were calculated in order to evidence between-groups and between-measures differences. To compare groups in the categorical variables, chi-square (χ_2_) was calculated, while the Mann–Whitney test was used to examine the possible differences between groups in the ordinal variables. 

The final phase of statistical analyses included calculation aimed to identify the predictors of PALs in the three testing waves. To identify the associations between the predictors and PAQ-A as a binomial criterion, with insufficient PALs coded as “1” and sufficient PALs as “2,” logistic regressions were calculated, with the odds ratios (ORs) and corresponding 95% CIs reported. However, since previous studies have regularly confirmed sport participation and male gender as being significantly associated with PALs in adolescence, before calculating the logistic regressions for the parental/familial predictors, we chanced associations between gender and sport participation with the PAL criteria and evidenced significant associations in our participants as well (please see later results). Accordingly, sport participation and male gender were included as covariates in the later logistic regressions calculated for the parental/familial variables as predictors of PALs in the three testing waves. Multivariate regression models were additionally calculated when more than one predictor was evidenced as being significantly related to the criterion. The model fit was checked using the Hosmer Lemeshow test (with a statistically significant χ_2_ indicating that the model did not adequately fit the data).

For all analysis, Statistica ver. 13.0 (Statsoft, Tulsa, OK, USA) was used and a *p*-value of 0.05 was taken as being statistically significant.

## 3. Results

The results of the factorial ANOVA with gender and measurement as the main effects and a gender × measurement interaction are presented in Table 1. The significant main effect for measurement evidenced that PALs significantly decreased over the course of the study (F = 94.17, *p* < 0.001). Meanwhile, the significant interaction effect (gender × measurement) pointed to differential changes in PALs for boys and girls (F = 6.61, *p* < 0.001). 

Figure 2 evidences the changes in PALs for boys and girls, as well as the significance of the post-hoc differences between and within genders. The boys had higher PALs than the girls at baseline, FU1, and FU2. The PALs of the girls decreased continuously from 16 to 18 years, with non-significant differences between 17 and 18 years. Meanwhile, the PALs among the boys decreased between baseline and FU1, but slightly increased from FU1 to FU2. 

As could be expected based on the previously presented ANOVA results, the boys were more likely to achieve sufficient PALs than the girls at baseline (OR = 5.84, 95% CI: 4.24–8.06), FU1 (OR = 4.55, 95% CI: 3.07–6.71), and FU2 (OR = 4.78, 95% CI: 3.31–6.89). Moreover, sport participation at baseline was a significant predictor of PALs in all three testing waves, with a lower likelihood of reaching appropriate PALs in adolescents who were not involved in sport (baseline: OR = 0.33, 95% CI: 0.27–0.40, FU1: 0.28, 95% CI: 0.22–0.37, FU2: 0.33, 95% CI: 0.26–0.41). Therefore, the logistic regressions calculated for the parental/familial variables as predictors of PALs were controlled for gender and baseline sport participation as covariates. 

The logistic regression calculated for the dichotomized PAL criterion at baseline (16 years of age), with gender and sport participation as covariates, evidenced better PALs among adolescents whose fathers were better educated (OR = 1.42, 95% CI: 1.17–1.84). Additionally, a lower likelihood of appropriate PALs was found in adolescents who reported a higher level of parental conflict (OR = 0.72, 95% CI: 0.58–0.88). The multivariate logistic regression, controlled for sport participation and gender, evidenced both predictors as being independently associated with PALs at baseline (paternal education: OR = 1.54, 95% CI: 1.23–1.94; conflict with parents/family: 0.69, 95% CI: 0.56–0.85). When logistic regression was used to calculate PALs at FU2 (17 years of age on average), no association was evidenced between the variables of parental education and PALs. Meanwhile, parental conflict remained significantly associated with the criterion, with a lower likelihood of being sufficiently active for adolescents who reported a higher level of conflict (OR = 0.83, 95% CI: 0.63–0.98). Finally, parental conflict was the single significant predictor of PALs at FU3 (OR = 0.75, 95% CI: 0.60–0.94) (Table 2). 

Additional data and univariate differences between groups based on physical activity levels at three testing waves are presented in Appendix A. 

## 4. Discussion

Our study aimed to prospectively examine the influence of parental/familial factors on PALs in adolescents aged 16–18 years. There are several important findings of this study. First, a significant decrease in PALs was evidenced during the observed period, and female gender was associated with a greater risk of being insufficiently active. Second, paternal education had a positive impact on children’s PALs only at baseline (i.e., when participants were 16 years of age), while such an influence was lost after the age of 17. Third, parental/familial conflict had a negative impact on PALs during the study period.

### 4.1. Gender and Changes in PALs

A decrease in PALs during adolescence has been evidenced globally [3,6]. Therefore, the decrease in PALs that we evidenced in adolescents herein was actually expected. However, this study, to some extent, fulfilled the picture regarding the trends in PAL changes during adolescence. In brief, when observing our results together with those previously reported using the same instruments, we can highlight the fact that the decrease in PALs in girls was actually continuous, at least in terms of the period between the age of 14 and 18 years. Specifically, Maric et al. recently evidenced a significant decrease in PALs between 14 and 16 years of age (first to third grades of high school) [3]. Meanwhile, the study of Stefan et al. indicated a decline between the second and third grades of high school [6], while in this study, we evidenced a decrease in PALs between the third and fourth grades. 

Collectively, this negative trend in PAL for the total sample (including boys and girls) can be explained by a great number of factors typical for this period of life, such as: (i) Changes in time requirements and life priorities (longer sitting time during school hours, during leisure activities, and at home), (ii) an increase in school duties, (iii) a stronger focus on academic achievement [31,32], and (iv) a decline in active transport [4]. Additionally, we can highlight quitting sport as one of the most important factors for the evidenced decrease in PALs found here. Namely, although we did not evidence the problem here, previous studies in the region evidenced large decrease in sport participation particularly for the late adolescence [33,34]. While an explanation of the different facets of the decline in PALs in adolescence is beyond the scope of this research, we instead focused on information provided directly throughout our investigation. 

Irrespective of the results which point to negative trends in PAL for total sample, when our results are combined with results of previous study done in Bosnia and Herzegovina trends in PALs for boys are inconsistent [3]. Most specifically we can highlight variable changes in PALs among adolescent boys from Bosnia and Herzegovina (2.42, 2.28, 2.75, 2.40, and 2.51 for 14-, 16-, 17- and 18-years of age, respectively) [3]. Possible explanations are provided in the following text. 

First, there is a certain possibility that selection of the participants influenced such results. Namely, in this study we included adolescents who finalized 4-year high-school education program (please see subsection Design and participants for details). The 4-year educational programs are known to be more challenging with regard to scholastic obligations than 3-year education programs, and children involved in 4-year programs are known to be better in educational achievements, including further educational plans (i.e., continuing education at College/University levels) [35]. While scholastic achievements are known to be associated with PALs, it is possible that initial selection of the study participants in this investigation resulted in specified discrepancies in trends of PALs to some extent, at least for boys. 

Second, the fact that PALs in boys slightly increases between the 2nd and 3rd testing wave may be a result of the fact that 2nd testing wave was organized at the end of the 3rd school year. Previous studies noted that this period is specific due to large drop-out from organized sports among boys [36], which consequently reduces even the PALs. In the following period (during the 4th year of high-school education), boys could re-organize their duties and interests, including initiation in some form of recreational physical exercise (in fitness and recreational centers), which could result even in increase of the PALs between 2nd and 3rd testing wave. 

A lower level of PALs in adolescent females than in boys is regularly reported [6,37]. Specifically for Southeastern Europe, a study conducted with students from Bosnia and Herzegovina indicated lower PALs among girls between 14 and 16 years of age [3]. Supportively, research conducted on adolescents from the territory of Croatia showed gender differences in the period between the third and fourth grades of high school, with higher PALs among boys [5]. Such differences are supported by the latest WHO data, stating that 84% of girls and 78% of adolescent boys do not meet the global PA level recommendations [38]. According to research to date, a few explanations are most commonly underlined as a cause of gender differences in PALs: (i) Social factors, (ii) attitudinal factors, and (iii) factors related to participation in organized sport [39].

When we talk about social factors in the context of gender differences, we are referring to the influence that the social environment (i.e., peers, teachers, and parents) has on PALs in adolescents [40]. More specifically, social factors—considering social–cultural and generational aspects—influence the modeling of PA involvement in males and females [41]. Specifically, PA through recreation and sport is often presented as an important element of masculinity in mass culture, resulting in higher social expectations toward men being physically active [41]. However, this actually means that girls often receive less support from the immediate environment related to participation in PA, while boys are exposed to a much larger number of social structures that encourage and influence PA and participation in PA [23,42]. This at least partially reflects the differences in PALs between genders during adolescence, resulting in higher PALs among boys. 

On the contrary, when we talk about the attitudes that affect PALs, we cannot ignore the simple fact that so-called “gender roles” define differences in attitudes, and therefore additionally contribute to the differences in PALs between adolescent boys and girls [43]. For instance, the competitive nature of sport is unappealing to a substantial number of today’s adolescents, but there is no doubt that such unattractiveness is more common in females. Girls therefore often feel incompetent/incapable of participating in sport and PA in general [31,44]. As a result, girls are generally less interested in PA, which certainly contributes to a their lower PALs in the end [45,46].

Finally, irrespective of the previously explained social and attitudinal factors related to PALs, we cannot ignore the sport factors that also directly contribute to differences in PALs between genders [43]. It cannot be ignored that PALs among children and adolescents mostly refers to organized sport in sport clubs, where the level of support and opportunity is not equal for both genders. Briefly, it is evident that the organized sport system, through organizational specifics, more accessible sport facilities, and more competition, undoubtedly favors males, leading to lower sport participation rates and much higher dropout rates in girls [47]. As direct support for previous discussion, we can highlight: (i) Higher levels of sport participation in boys than in girls observed herein (please see Appendix A for details), and (ii) a significant influence of sport participation on the PALs in all three testing waves. 

### 4.2. Parental Education and PALs

Physically active parents inspire their children to develop healthy physical behaviors [48]. Logically, parental influence is one of the primary factors influencing the behavior of young people, and thus the habits associated with PALs [12,18,19]. Accordingly, a large number of studies to date have examined familial and parental influence on PALs in children and adolescents [13,49]. However, the results are inconsistent. While some studies have indicated a positive impact of higher parental education on children’s PALs, others show conflicting results [5,24,44]. This is consistent with our results, given that: (i) Maternal education was not identified as a factor influencing PALs, and (ii) the influence of paternal education on PALs was significant only at the study baseline. Interestingly, the positive influence of paternal education on PALs at baseline (when participants were 16 years of age) is consistent with a recent report where Maric et al. highlighted a positive relationship between same variables in adolescents over the period of 14–16 years of age [3]. The background of this relationship is discussed below. 

Even though studies show conflicting findings when it comes to maternal or paternal influence on PAL and sports involvement, we may assume that our finding is in accordance with studies highlighting fathers as one of the most important factors influencing PALs and involvement in sport [50,51]. Therefore, we can assume that the significant impact of (exclusively) paternal education on PALs at baseline is partly due to the greater impact they generally have on the PALs of their children. The reason for this relationship is probably based on the social perception that men are more competent in terms of PA and sport than women. The authors of the current study are of the opinion that this is particularly noticeable in the area of Southeastern Europe, where sport represents one of the most common topics of socializing in male environments [52]. However, it is also possible that fathers with a higher educational level, knowing the benefits of PA, actively encourage their children to participate in PA and sport, as it has been recently suggested [3]. However, for the moment, we cannot clearly define which of the factors mentioned above significantly affects the established correlation at baseline; hence, it should be studied in more detail in the future.

To the best of our knowledge, this is the first study to report the time points at which parental education is no longer a factor of influence on children’s PALs. Evidently, the age of 17 years should be considered a certain “critical point,” at least for adolescents from Bosnia and Herzegovina. The authors believe that the reason for this is related to maturation, which similarly affects PALs, but also the sociocultural determinants of PALs. More specifically, it is assumed that through this period, parental supportive behavior decreases in accordance with the maturation of adolescents. During the reported period, adolescents go through social change and the opinions of their peers become far more important than the opinions of their family members [24,53]. 

At the same time, the perceptions of adolescents themselves change accordingly [24]. Thereby, previously appreciated parental influence changes in the eyes of adolescents. The assumption is that parental input creates repulsion, since adolescents feel mature enough to make their own decisions and form their own conclusions about individual behaviors, which they could feel deprived of because of parental interference [24]. Adolescents strive for emotional autonomy from their parents so that they can have the opportunity to develop into an independent person [54]. Although further studies are needed in order to examine this issue in more detail, we may assume that the previously discussed information at least partially explains the lack of influence of parental education on PALs after 17 years of age. Moreover, this probably explains the non-significant association between parental monitoring and parental care with PALs in the studied adolescents herein. 

### 4.3. Parental Conflict and PALs

Parental/familial conflict evidenced at baseline had a negative impact on PALs during the whole study period. Interestingly, a recent study conducted in the same country showed no association between the same parental/familial variables and PALs in younger adolescents (12–14 years of age) [3]. It is particularly important to note that the previous study used identical measurement tools to those used here. Therefore, it seems that such relationships and the possible influence of parental/familial conflict on children’s PALs are characteristic of older adolescents (e.g., from 16 to 18 years of age). Explanations should be sought for the possible causes of the conflict itself. 

Older adolescence is marked by a series of rebellious behavior [55]. Adolescents of such an age are often prone to thinking that they know and understand more than they objectively do, trying new things, taking risks, and testing boundaries [56]. Therefore, it is possible that the conflict between children and parents/family is a result of too directive and/or restrictive parental behavior. This would naturally jeopardize the autonomy of adolescents, which is known to be extremely important during this period [24]. Altogether, this may result in the rebellion of children against all “pro-social behaviors,” including those associated with PALs (i.e., participation in sport, active transportation, and recreational activities with family and friends). 

Supportively, previous studies have indicated that controlling and directive parental behavior in some cases have a negative impact on adolescents’ PALs. For instance, Wing et al., in a sample of 594 adolescents from Canada, pointed to the negative impact of parental control on self-efficacy beliefs and youth enjoyment of PA, which the authors highlighted as the two main determinants of PA-related behavior [57]. Therefore, it is important to take into account the fact that conflicts between parents and adolescents are one of the most characteristic/common features of adolescence. Therefore, although such conflict is most often characterized by mild arguments, disagreements, and conflicts over everyday issues, it is necessary to pay attention to the negative effects that these conflicts can have on young people’s behavior, including the behavior related to PA.

### 4.4. Limitations and Strengths

This study has several limitations. First, physical activity was not directly measured but was evidenced by self-administered questionnaires. However, knowing that the PAQ-A has been repeatedly validated in similar samples of participants, we believe that the results obtained herein are plausible. Furthermore, this study did not observe some of the important covariates of the predictors and outcomes (i.e., scholastic factors and physical literacy), which limits the possibility to discuss the established relationships more profoundly. Third, we observed participants from one specific country and location; therefore, the results are generalizable only to similar regions. 

This study is one of the first where PA was observed across a relatively short time span off approximately six months. Therefore, it allowed us to better understand the complex and dynamic relationships between the parental variables and physical activity among adolescents. Additionally, this study is a continuation of previous research conducted with younger participants from the same country. Such an experimental approach, together with a prospective design, allowed us to identify and explain the trends in PALs changes, as well as to discuss the relationships between the studied predictors and the PALs from 14 to 18 years of age. 

## 5. Conclusions

The results of this study point to a significant decline in PALs in the period between 16 and 18 years of age. Knowing that previous studies conducted in the same region have evidenced a similar decline in the previous two years, we may highlight the continuous decline in PALs from 14 to 18 years Therefore, in order to prevent a decline in PALs in this period of life, public health authorities should target all ages with similar respect and attention. In other words, there is no evidence that a specific age only should be considered as “critical” in terms of a more rapid decline in PALs. 

However, there are certain evidences that negative trends in PALs among adolescent boys are not continuous between the age of 14 and 18. There is certain possibility that changes in sport participation (i.e., quitting organized competitive sports, involvement in fitness/recreational physical exercising) during adolescence influence such results. Therefore, this issue should be more precisely studied in future investigations. 

This study also evidenced a significant influence of paternal education level on children’s PALs at the age of 16, with no association between parental education and children’s PALs after the age of 17 years. This information is important from the perspective of the development of a targeted intervention aimed at the prevention of a decline in PALs. Namely, while children whose parents (fathers) are better educated should be observed as being “protected” against a decline in PALs until the age of 16 years, this “protective mechanism” seems to disappear in older adolescence. 

Contrary to the association between parental education and PALs that exists in younger adolescence, this study brought evidence of parental conflict as a significant risk factor of lower PALs later in adolescence (i.e., >17 years of age). As a result, when targeting older adolescents at risk of lower PALs, special attention should be placed on those who self-report higher parental/familial conflict. Meanwhile, it seems that factors of parental conflict do not correlate with PALs in younger adolescence. 

Lastly, this study evidenced differential dynamics of the changes in PALs between boys and girls in the observed period. Specifically, while the PALs among the girls decreased continuously, the boys experienced variable changes in PALs, with a decrease between 16 and 17 years and a slight increase between 17 and 18 years of age. Therefore, further analysis is needed in order to better understand such differences between and within genders. 

## Figures and Tables

**Figure 1 healthcare-09-00132-f001:**
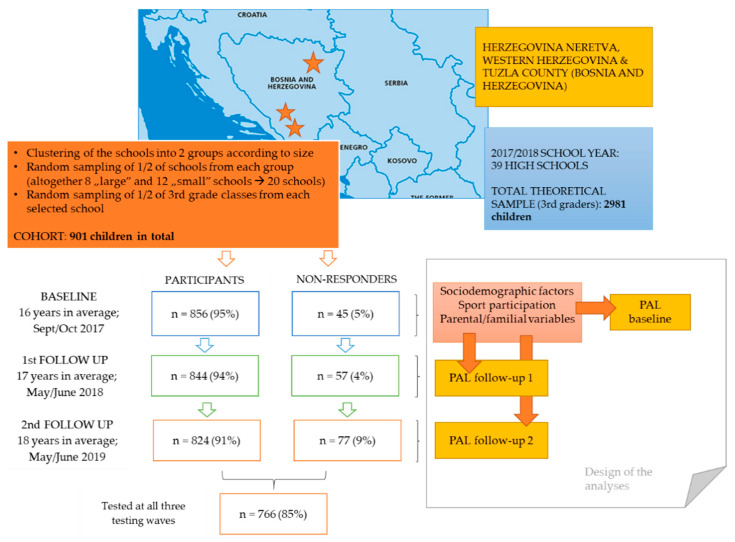
Study location, design, and testing sequences.

**Figure 2 healthcare-09-00132-f002:**
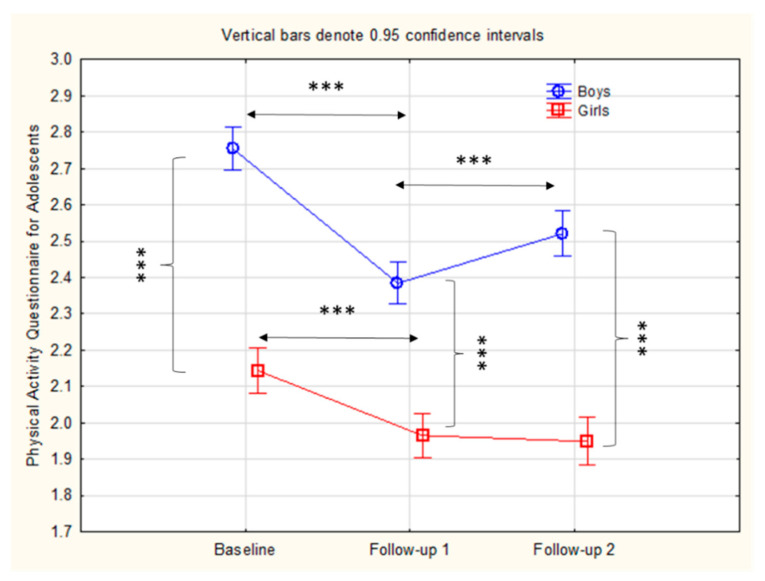
Descriptive statistics (Mean ± 95% CI) and statistical significance of the differences between and within groups/genders obtained by post-hoc Schefee test (*** indicates *p-level* of 0.001).

**Table 1 healthcare-09-00132-t001:** Results of the factorial analysis of variance for repeated measurements.

Variable	Main Effects				Interaction	
	Gender		Measurement		Gender × Measurement	
	F-test	*p*-level	F-test	*p*	F-test	*p*-level
PAQ-A	94.17	0.001	83.05	0.001	6.61	0.001

**Table 2 healthcare-09-00132-t002:** Correlates of physical activity levels at baseline, and two follow-up measurements; results of the logistic regression analysis controlled for gender and sport participation as covariates (N = 766).

Variables	PAL Baseline		PAL Follow-Up 1		PAL Follow-Up 2	
	OR	95% CI	OR	95% CI	OR	95% CI
Socioeconomic status	0.79	0.44–1.41	0.89	0.48–1.67	0.65	0.35–1.21
Paternal education	1.42	1.17–1.84	1.16	0.98–1.49	1.23	0.97–1.56
Maternal education	1.16	0.95–1.43	1.18	0.88–1.41	0.90	0.71–1.13
Parental/familial conflict	0.72	0.58–0.88	0.83	0.63–0.98	0.75	0.60–0.94
Parental absence	1.13	0.97–1.32	1.06	0.88–1.28	1.14	0.96–1.36
Parental care	0.80	0.62–1.04	1.09	0.81–1.47	1.01	0.76–1.33
Parental questioning	1.16	0.93–1.44	1.09	0.87–1.38	1.20	0.98–1.47

## Data Availability

Data is available here: https://www.dropbox.com/s/u48vaisp9wgyifn/data%20file.xlsx?dl=0.

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
