# Peer review of "Familial and Parental Predictors of Physical Activity in Late Adolescence: Prospective Analysis over a Two-Year Period"

_healthcare, 2021, doi:10.3390/healthcare9020132_

Round 1

Reviewer 1 Report

Dear Authors,

Your study in of interest and well-constructed. We have few observations that we hope will be useful for your publication.

Page 1 you write:

  1. Introduction

“Despite the globally known benefits of physical activity (PA) and the negative health outcomes of an inadequate physical activity level (PAL)”  PAL can be associated by the future lecturer with inadequate physical activity level, then to avoid confusion, we suggest: “of an inadequate PAL (physical activity level)”

How can you explain that in you study with 14-16 years adolescent (22) medium PAL of boys was 2.42 to 2.28 with 61 to 68% of boys with insufficient PAL, and in the present study the 16y boys obtained in medium 2.75 that can be considered sufficient PAL, reaching at FU1 2.4 and at FU2 ~2.51. On the basis of those results can you confirm that in boys there is a continuous decrease of PAL, like in girls? In girls using both studies it looks like a continuous decrease of PAL during early and latter adolescence 2.17, 2.01, ~2.15, ~1.98, ~1.96, but in boys it is not so clear, in addition, the last year’s levels of PAL are superior than the first years. Our impression is that what is true for girls and perhaps the whole group of teens, is not true for the boys.  We suggest that you reflect on that and discuss this in gender discussion, eventually reflecting the news insides in the conclusions.

Page 13, you write: “Fathers are often considered as one of the most important factors influencing PALs and involvement in sport [47,48]. Therefore, we can assume that the significant impact of (exclusively) paternal education on PALs at baseline is partly due to the greater impact they generally have on the PALs of their children. The reason for this relationship is prob-ably based on the social perception that men are more competent in terms of PA and sport than women. The authors of the current study are of the opinion that this is particularly noticeable in the area of Southeastern Europe, where sport represents one of the most common topics of socializing in male environments [49].” We agree with your final opinion, but not with the beginning affirmation about fathers, because in some other countries / studies, mothers are considered equal or more (in children) influent than fathers. We suggest you discuss better this part and also consider that the literature is not uniform about fathers and mothers influence, probably depending on the culture.

You write: “Therefore, it seems that such relationships and the possible influence of parental/familial conflict on children’s PALs are characteristic of older adolescents (e.g., from 16 to 18 years of age).” Another explanation for the conflict with parent influence on PAL can be reaction against the earlier sports practise excess on the insistence of the parents in children time and early adolescence. The decrease of sport practise in your study can also be discussed on the light of studies about parent’s behaviour and teens Drop out of sports.

Author Response

Dear Authors,

Your study in of interest and well-constructed. We have few observations that we hope will be useful for your publication.

RESPONSE: Thank You for recognizing the value of our idea, and for providing additional comments and suggestions. We tried to follow it and amended the manuscript accordingly. Please see below for more details on changes and improvements. Staying at your disposal.

Introduction

“Despite the globally known benefits of physical activity (PA) and the negative health outcomes of an inadequate physical activity level (PAL)” PAL can be associated by the future lecturer with inadequate physical activity level, then to avoid confusion, we suggest: “of an inadequate PAL (physical activity level)”

RESPONSE: Thank you for this suggestion. Text is amended accordingly. Text reads: “Despite the globally known benefits of physical activity (PA) and the negative health outcomes of an inadequate (PAL) physical activity level, physical inactivity indicates a trend of pandemic proportions, with no improvement since 2001 [1].” (Please see first paragraph of the Introduction)

How can you explain that in you study with 14-16 years adolescent (22) medium PAL of boys was 2.42 to 2.28 with 61 to 68% of boys with insufficient PAL, and in the present study the 16y boys obtained in medium 2.75 that can be considered sufficient PAL, reaching at FU1 2.4 and at FU2 ~2.51. On the basis of those results can you confirm that in boys there is a continuous decrease of PAL, like in girls? In girls using both studies it looks like a continuous decrease of PAL during early and latter adolescence 2.17, 2.01, ~2.15, ~1.98, ~1.96, but in boys it is not so clear, in addition, the last year’s levels of PAL are superior than the first years. Our impression is that what is true for girls and perhaps the whole group of teens, is not true for the boys.  We suggest that you reflect on that and discuss this in gender discussion, eventually reflecting the news insides in the conclusions.

RESPONSE: Thank you for noticing it. We provided some possible explanations about it in the discussion and conclusion section.

In Discussion, text reads:

“Irrespective of the results which point to negative trends in PAL for total sample, when our results are combined with results of previous study done in Bosnia and Herzegovina trends in PALs for boys are inconsistent [3]. Most specifically we can highlight variable changes in PALs among adolescent boys from Bosnia and Herzegovina (2.42, 2.28, 2.75, 2.40, and 2.51 for 14-, 16-, 17- and 18-years of age, respectively) [3]. Possible explanations are discussed in the following text.

First, there is a certain possibility that selection of the participants influences such results. Namely, in this study we included adolescents who finalized 4-year high-school education program (please see subsection Design and participants for details). The 4-year educational programs are known to be more challenging with regard to scholastic obligations than 3-year education programs, and children involved in 4-year programs are known to be better in educational achievements, including further educational plans (i.e. continuing education at College/University levels). Therefore, it is possible that initial se-lection of the study participants in this investigation resulted in specified discrepancies in trends of PALs to some extent, at least for boys.

Second, the fact that PALs in boys slightly increase between the 2nd and 3rd testing wave (from 2.40 to 2.51) may be a result of the fact that 2nd testing wave was organized at the end of the 3rd school year. Previous studies noted that this period is specific due to large drop-out from organized sports [35], which consequently reduces even the PALs. In the following period (during the 4th year of high-school education), boys could re-organize their duties and interests, including initiation in some form of recreational physical exercise (in fitness and recreational centers), which could result even in increase of the PALs between 2nd and 3rd testing wave.”

Please see highlighted text in Discussion section.

Also, it is pointed in the Conclusion section, and text now reads: “However, there are certain evidences that negative trends in PALs among adolescent boys are not continuous between the age of 14 and 18. There is certain possibility that changes in sport participation (i.e. quitting organized competitive sports, involvement in fitness/recreational physical exercising) during adolescence influence such results. There-fore, this issue should be more precisely studied in future investigations. “

(please see  highlighted text in Conclusion section).

Page 13, you write: “Fathers are often considered as one of the most important factors influencing PALs and involvement in sport [47,48]. Therefore, we can assume that the significant impact of (exclusively) paternal education on PALs at baseline is partly due to the greater impact they generally have on the PALs of their children. The reason for this relationship is probably based on the social perception that men are more competent in terms of PA and sport than women. The authors of the current study are of the opinion that this is particularly noticeable in the area of Southeastern Europe, where sport represents one of the most common topics of socializing in male environments [49].” We agree with your final opinion, but not with the beginning affirmation about fathers, because in some other countries / studies, mothers are considered equal or more (in children) influent than fathers. We suggest you discuss better this part and also consider that the literature is not uniform about fathers and mothers influence, probably depending on the culture.

RESPONSE: Thank you for your comment. Text is amended accordingly. Text reads: “Even though studies show conflicting findings when it comes to maternal or paternal influence on PALs and sports involvement, we may assume that our finding is in accordance with studies highlighting fathers as one of the most important factors influencing PALs and involvement in sport [47,48].” (Please see second paragraph of subheading 4.2; Thank you.)

You write: “Therefore, it seems that such relationships and the possible influence of parental/familial conflict on children’s PALs are characteristic of older adolescents (e.g., from 16 to 18 years of age).” Another explanation for the conflict with parent influence on PAL can be reaction against the earlier sports practise excess on the insistence of the parents in children time and early adolescence. The decrease of sport practise in your study can also be discussed on the light of studies about parent’s behaviour and teens Drop out of sports.

RESPONSE: Thank you for your suggestion, authors will certainly have this in mind in future research, however since we did not discuss PA changes in the context of sport factors we decided not to mention sport participation in this context as it is not presented in our work, and it is not supported by implemented statistical analysis. However, according to one of your previous suggestions, the “problem” of sport drop-out is accentuated in the Conclusion section as being important issue which should be studied in details in further studies.  

Staying at your disposal

Authors

Reviewer 2 Report

Familial and Parental Predictors of Physical Activity in Late Adolescence: Prospective Analysis Over a Two-Year Period

Thanks for this manuscript, I appreciate the opportunity to review it. The study aimed to prospectively evaluate the parental/familial factors associated with physical activity levels (PALs) among older adolescents.

Minor reviews:

Please, use either physical activity level (PAL) or physical activity levels (PALs).

Introduction

Paragraph 1. A previous study (Guthold et al., 2020) showed a decrease in the prevalence of insufficient physical activity between 2001 and 2016 among boys; however, these changes were not significant among girls. I think that the authors may turn this point clearer. Moreover, the authors may point out that even with reduction in insufficient physical activity among boys, the majority of adolescent does not reach the international PA guidelines.

Guthold, Regina, et al. "Global trends in insufficient physical activity among adolescents: a pooled analysis of 298 population-based surveys with 1· 6 million participants." The Lancet Child & Adolescent Health 4.1 (2020): 23-35.

In general, the introduction is well presented. I suggest just one more point: although the authors have reported the international levels of physical activity, I suggest including information about national/regional physical (in)activity level as well.

Table 2. The table 2 could benefit with the inclusion of the number of observations in each model (baseline; follow-up 1; follow-up 2).

Table 2. Regarding to the association between parental/ familiar conflict and PAL follow-up 2, the OR value was [0.75 (0.6; 0.94)]. To avoid any missing value, please, confirm whether “0.6” really means “0.60”. I suggest it because the authors reported a OR of “0.80” regard of the association between parental care and PAL baseline.

Author Response

Suggestions for Authors

Familial and Parental Predictors of Physical Activity in Late Adolescence: Prospective Analysis Over a Two-Year Period

Thanks for this manuscript, I appreciate the opportunity to review it. The study aimed to prospectively evaluate the parental/familial factors associated with physical activity levels (PALs) among older adolescents.

RESPONSE: Thank you for your support and review. We followed your suggestions and amended manuscript accordingly. For more details, please see responses below. Staying at your disposal.

Minor reviews:

Please, use either physical activity level (PAL) or physical activity levels (PALs).

RESPONSE: Corrected. Thank you

Introduction

Paragraph 1. A previous study (Guthold et al., 2020) showed a decrease in the prevalence of insufficient physical activity between 2001 and 2016 among boys; however, these changes were not significant among girls. I think that the authors may turn this point clearer. Moreover, the authors may point out that even with reduction in insufficient physical activity among boys, the majority of adolescent does not reach the international PA guidelines.

Guthold, Regina, et al. "Global trends in insufficient physical activity among adolescents: a pooled analysis of 298 population-based surveys with 1· 6 million participants." The Lancet Child & Adolescent Health 4.1 (2020): 23-35.

RESPONSE: Thank you for this suggestion. Text is amended accordingly. Text reads: “Despite the globally known benefits of physical activity (PA) and the negative health outcomes of an inadequate (PALs) physical activity levels, physical inactivity indicates a trend of pandemic proportions, with almost no improvement since 2001 [1]. Guthold et al. reported a slight decrease in boys insufficient PA since 2001, while there were no changes over time in girls, and in the general adolescent population [2].” (Please see first paragraph of the Introduction)

In general, the introduction is well presented. I suggest just one more point: although the authors have reported the international levels of physical activity, I suggest including information about national/regional physical (in)activity level as well.

RESPONSE: Thank you for your suggestion, text is amended accordingly and reads: "Trends of PA decline in adolescence are repeatedly highlighting in Southeastern Europe as well [3-6].” (Please see first paragraph of the Introduction)

Table 2. The table 2 could benefit with the inclusion of the number of observations in each model (baseline; follow-up 1; follow-up 2).

RESPONSE: Thank you for this suggestion. Number of observations is presented in Table title. Note that in all three models we included only those adolescents who were tested at each single testing wave (N = 766)

Table 2. Regarding to the association between parental/ familiar conflict and PAL follow-up 2, the OR value was [0.75 (0.6; 0.94)]. To avoid any missing value, please, confirm whether “0.6” really means “0.60”. I suggest it because the authors reported a OR of “0.80” regard of the association between parental care and PAL baseline.

RESPONSE: Thank you for noticing, it was typewriting mistake, result in Table 2. was amended accordingly, and value now reads “0.60”. (Please see results, Table 2.)

Staying at your disposal

Authors